
# A general definition of $JT_a$ – deformed QFTs

Tarek Anous[1,2*] and Monica Guica[3,4]

**1** $\Delta$-Institute for Theoretical Physics & Institute for Theoretical Physics
University of Amsterdam, Science Park 904, 1090 GL, Amsterdam, The Netherlands
**2** Department of Physics and Astronomy, University of British Columbia,
Vancouver, BC V6T 1Z1, Canada
**3** Institut de Physique Théorique, CEA Saclay, CNRS, 91191 Gif-sur-Yvette, France
**4** Department of Physics, Stockholm University, AlbaNova, 106 91 Stockholm, Sweden

* t.m.anous@uva.nl

## Abstract

We propose a general path-integral definition of two-dimensional quantum field theories deformed by an integrable, irrelevant vector operator constructed from the components of the stress tensor and those of a $U(1)$ current. The deformed theory is obtained by coupling the original QFT to a flat dynamical gauge field and "half" a flat dynamical vielbein. The resulting partition function is shown to satisfy a geometric flow equation, which perfectly reproduces the flow equations for the deformed energy levels that were previously derived in the literature. The S-matrix of the deformed QFT differs from the original S-matrix only by an overall phase factor that depends on the charges and momenta of the external particles, thus supporting the conjecture that such QFTs are UV complete, although intrinsically non-local. For the special case of an integrable QFT, we check that this phase factor precisely reproduces the change in the finite-size spectrum via the Thermodynamic Bethe Ansatz equations.

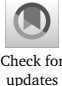

# 1   Introduction

One of the simplest ways to conceive of a UV-complete QFT is to imagine the existence of a UV CFT fixed point, which is then deformed by a relevant operator. The deformation triggers a renormalization group flow, at some point along which lies the QFT under study. However, it is clear that far from all UV-complete QFTs of interest are described within this framework - most notably theories of quantum gravity, but also, for example, non-commutative [1] and dipole theories [2,3]. It is thus of great interest to better understand other possible ultraviolet behaviours in QFT.

A very simple example of a UV-complete QFT that does not fall within the usual framework is the worldsheet theory of a bosonic string in static gauge, described by the Nambu-Goto action. The S-matrix of this theory was computed in [4,5] using integrability techniques and is given by a simple momentum-dependent phase, which is obtained from the trivial S-matrix by a large gauge transformation that aligns the worldsheet coordinates to the target space ones. The ultraviolet behavior of this S-matrix is incompatible with the existence of a local UV CFT fixed point; rather, it was argued to describe gravitational scattering on the worldsheet. It was later shown [6] that dressing the S-matrix of *any* UV-complete two-dimensional QFT with this gravitational phase turns it into a consistent S-matrix for a UV-complete theory that lacks a conventional CFT fixed point in the UV. However, at the time, no action principle was specified for these QFTs.

A different formulation of what turned out to be the same theories, this time based on an action principle, was put forth in [7,8] and consists of gradually deforming any translationally-invariant QFT by the so-called $T\bar{T}$ operator [9]: a *universal,* irrelevant, bilinear operator constructed from the components of the stress tensor. The very special properties enjoyed by this operator ensure that certain observables, such as the energy levels $E_n$ of the theory in finite volume, can be solved exactly as a function of the perturbative $T\bar{T}$ coupling, $\mu$. Moreover, if the QFT is integrable, the S-matrix is only changed by a Castillejo-Dalitz-Dyson (CDD) phase factor [8] that coincides with the gravitational dressing factor of [6]. Note, however, that this definition is intrinsically perturbative, valid only at length scales much larger than the one set by the dimensionful coupling $\mu$, and the potential UV-completeness of the deformed theory is completely obscured in this picture.

Soon afterwards, [10] provided a framework that unifies the exact solubility of the irrelevant $T\bar{T}$ deformation of general QFTs with the gravitational dressing procedure of the S-matrix. This was achieved by reformulating the deformation as coupling the original QFT to a topological theory of gravity, in which the metric is allowed to fluctuate, but in a way that forces it to be everywhere flat. The physical effect of this coupling is to implement a dynamical change of coordinates that is characteristic of $T\bar{T}$ deformations [11] and is the direct analogue of the gauge transformation from conformal to static gauge on the bosonic string worldsheet. This picture recovers both the $T\bar{T}$ deformation at small coupling and the full S-matrix dressing at arbitrary coupling.

By computing the Euclidean path integral over fluctuations about the flat metric, Cardy

subsequently showed [12] that the torus partition function of $T\bar{T}$-deformed QFTs obeys a simple flow equation

$$\frac{\partial \Psi}{\partial \mu} = \frac{1}{4} \epsilon^{ab} \epsilon_{\alpha\beta} \frac{\partial^2 \Psi}{\partial L_\alpha^a \partial L_\beta^b}, \tag{1.1}$$

where the $L_\alpha^a$ parametrize the lengths of the torus, with area $\mathcal{A}$, and $\Psi \equiv Z_{T\bar{T}}/\mathcal{A}$. When expressed in terms of the energy levels $E_n$, this flow equation reproduces the previously known results on the spectrum. The final check that the non-perturbative definition of the $T\bar{T}$ deformation by coupling to topological gravity does reproduce the perturbative $T\bar{T}$ deformation, was the calculation [13] of the exact torus partition function of the theory, which showed that it satisfies precisely the equation (1.1).

The study of the $T\bar{T}$ deformation has been an active field of research over the past few years (see [14–20] for a selection of the developments). However, as shown in [7], any irrelevant deformation constructed from an antisymmetric combination of the components of two conserved currents is integrable, in the same way as the $T\bar{T}$ deformation. A deformation that shares the universality properties of $T\bar{T}$ is the so-called $J\bar{T}$ deformation [21], constructed from the components of a global conserved $U(1)$ current and those of the stress tensor associated to $\bar{z}$ translations. The $J\bar{T}$ deformation was originally introduced as an $SL(2,\mathbb{R})$-preserving irrelevant deformation of two-dimensional CFTs, destined to model the behaviour of holographic duals to extremal black holes[1] [23, 24]. The holographic dictionary for $J\bar{T}$-deformed CFTs was worked out in [25]. A very interesting single-trace version of the deformation was proposed in [26,27]. The modular properties of the partition function were studied in [28], while correlation functions were studied in [29]. The effect of the deformation on the S-matrix of integrable QFTs was recently studied in [30].

In this article, conformal symmetry will not play any role. Consequently, we replace $\bar{T}$ by the generator $T^\alpha{}_a$ of translations along any chosen direction $x^a$. We henceforth call these "$JT_a$-deformed QFTs," and define the deformation infinitesimally via

$$\frac{\partial S_{JT_a}}{\partial \mu^a} = \int d^2\sigma \, e \, T^\alpha{}_a \, \epsilon_{\alpha\beta} J^\beta, \tag{1.2}$$

where the coupling constant $\mu^a$ is a vector of *fixed* direction, but varying amplitude. This definition is taken to hold in both Euclidean and Lorentzian signatures, with the Euclidean and Lorentzian couplings related via an appropriate analytic continuation. The flow equations for the $JT_a$-deformed energy levels were first understood in [19], via a method based on coupling to background fields. Similar flow equations were obtained in [20] using uniform lightcone gauge. For large enough values of $|\mu^a|$, the energies tend to become imaginary, raising questions about the existence of these theories all the way up into the UV.[2]

The main goal of this article is to provide a general, non-perturbative definition of $JT_a$-deformed QFTs that simultaneously makes manifest their UV completeness and correctly reproduces the deformed spectrum, similar to what the coupling to topological gravity achieved for $T\bar{T}$. As we already reviewed, in the $T\bar{T}$ case this coupling was mainly designed to implement the dynamical change of coordinates between the analogues of conformal and static gauge. It turns out that in the case of $J\bar{T}$, the effects of the deformation are well captured by a combination of a gauge transformation and a coordinate change that only affects half of the components of the vielbein [25]. It is thus natural to propose a path integral definition in which we couple the original QFT to a flat external gauge field and "half" a flat dynamical

---

[1]For the most recent developments in this direction, see [22].

[2]Our point of view on this issue is that the imaginary energies may be just be an artifact of (illegally) putting the theory at finite volume, see [31] for relevant comments.

vielbein.[3]

As a first check of our proposal, we compute the partition function of the deformed QFT which, just like in $T\bar{T}$, is one-loop exact and turns out to satisfy an intrinsically geometric flow equation

$$\frac{\partial \Psi}{\partial \mu^a} = \epsilon_{\alpha\beta} \frac{\partial^2 \Psi}{\partial L_\alpha{}^a \partial \nu_\beta}, \tag{1.3}$$

where $\Psi = Z_{JT_a}/\mathcal{A}$ and the direction of $\mu^a$ is held fixed. Upon a proper identification of the parameters, the resulting flow equations for the energy levels perfectly match the literature, thus confirming that our proposal is correct.

Our general definition also allows us to compute the S-matrix dressing of $JT_a$-deformed QFTs, by an analogous calculation to that performed in [10] for $T\bar{T}$. As expected, the deformed S-matrix only differs from the original one by a phase, schematically given by

$$S_{\mu^a}(p_i^a, q_i) = e^{-i\mu_a \sum_{i<j}\left(q_i p_j^a - q_j p_i^a\right)} S_0(p_i^a, q_i), \tag{1.4}$$

where $S_0(p_i^a, q_i)$ is the S-matrix of the undeformed QFT, $p_i^a$ is the momentum vector of the $i^{\text{th}}$ particle, $q_i$ is the its $U(1)$ charge, and the incoming (outgoing) particles are (anti-)ordered according to their rapidities. Assuming $S_0$ describes a QFT with a CFT fixed point in the UV, the exact expression above indicates that the deformed theory is also UV-complete, but no longer possesses a conventional UV fixed point.

As a final check of our proposal, we show explicitly that in integrable theories, the additional dressing factor (1.4) of the scattering matrix precisely reproduces the change in the finite-size spectrum via the Thermodynamic Bethe Ansatz (TBA) equations. This confirms that the two predictions that follow from our general definition of $JT_a$-deformed QFTs are consistent with each other.

This paper is organized as follows. In section 2 we present the general definition of the $JT_a$ deformation in terms of coupling to topological background fields. In section 3, we evaluate the partition function, show that it satisfies the flow equation (1.3), and compare our results to the literature. We also reproduce the explicit solutions to this flow equation for generic $\mu^a$. In section 4 we compute the S-matrix dressing due to the $JT_a$ deformation and, for the case of integrable QFTs, we relate this dressing factor to the finite-size spectrum derived in section 3 via the TBA equations.

## 2 General definition of $JT_a$ - deformed QFTs

We consider two-dimensional quantum field theories deformed by an irrelevant Smirnov-Zamolodchikov [7] operator constructed from the components of a $U(1)$ current $J^\alpha$ and those of the current associated to translations in some fixed direction $x^a$, i.e. the stress tensor $T^\alpha{}_a$. The deformation is defined infinitesimally in (1.2), where $\mu^a$ is a coupling that points in a *fixed* direction of our choosing.

Since the deforming operator carries spin, the deformation breaks Lorentz invariance, implying that the stress tensor is not symmetric. However, it is conserved: $\nabla_\alpha T^\alpha{}_a = 0$. While the deformation (1.2) is a priori defined on flat space, its generalization to curved space is simple [25]: Greek indices are promoted to spacetime indices, while Latin ones indicate tangent

---

[3]Recently, [32] studied a path integral realisation of joint $J\bar{T}$, $T\bar{J}$ and $T\bar{T}$ deformations, where $J$ and $\bar{J}$ are independent currents, in terms of coupling to a *full* topological vielbein and *two* independent gauge fields. This kernel is different from ours - roughly, it is a doubling of what we find - and it reflects the fact that in the proposal of [32] it is impossible to turn off one of the $J\bar{T}$ or $T\bar{J}$ couplings in the intermediate steps of the calculations. We comment more on the relation to our proposal in footnote 6.

space directions. In particular, the coupling $\mu^a$ is a tangent space vector. The $J\bar{T}$ deformation of [21] corresponds to the particular choice of $\mu^a$ null.

The above definition holds in Lorentzian signature, where Hermiticity of the deforming operator requires that $\mu^a$ be real. Upon moving to Euclidean signature by continuing $t \to -i\tau$, the Euclidean time component of the deformation remains real, $\mu_E^\tau = \mu^t$ in our conventions (namely, $\epsilon^{t\phi} = \epsilon^{\tau\phi}$); however, the space component must be taken to be purely imaginary, $\mu_E^\phi = -i\mu^\phi$, in order to preserve the reality of the Euclidean action.

We would now like to present a way to define the above $JT_a$-deformed QFTs in a generic fashion by coupling to background fields, following the work [13] on $T\bar{T}$, which we review below. In the rest of this paper, we borrow the terminology suggested in the bosonic string picture, by referring to the background spacetime of the original QFT as the 'worldsheet,' and we will call the space parametrized by the $X^a$ coordinates (defined below) the 'target space.'

## 2.1 Brief review of the $T\bar{T}$ deformation as coupling to topological gravity

The $T\bar{T}$ deformation can be non-perturbatively defined by a particular coupling of the original quantum field theory to a special theory of topological gravity

$$S_{T\bar{T}} = \int d^2\sigma\, e\left[\mathcal{L}_{\mathrm{QFT}}(\varphi, e_\alpha{}^a) - \frac{1}{2\mu}\epsilon^{\alpha\beta}\epsilon_{ab}(\partial_\alpha X^a - e_\alpha{}^a)(\partial_\beta X^b - e_\beta{}^b)\right], \qquad (2.1)$$

where $e_\alpha{}^a$ is a nontrivial background vielbein. The $X^a$ are a pair of auxiliary fields whose equations of motion set $\partial_{[\alpha} e_{\beta]}{}^a = 0$, implying that the curvature is exactly zero. This is important for the quantum theory and is ultimately responsible for the solubility of this deformation, as the semiclassical sum over topologies only receives a contribution from the torus, with the higher-genus Riemann surfaces being precluded by the Gauss-Bonnet theorem.

A simple way to motivate this proposal is by following [12]. The definition of the $T\bar{T}$ deformation on flat space is

$$\frac{\partial S_{T\bar{T}}}{\partial \mu} = -\frac{1}{2}\int d^2\sigma\, e\, \epsilon_{\alpha\beta}\, \epsilon^{ab}\, T^\alpha{}_a T^\beta{}_b\,. \qquad (2.2)$$

A standard way to deal with composite operators is to use the Hubbard-Stratonovich trick, which in this case amounts to coupling the original QFT to a dynamical metric. However, as shown in [12], the conservation of the stress tensor implies that the path integral reduces to one only over flat metrics, at least infinitesimally. When passing from metrics to vielbeine, a simple way to enforce this constraint is to introduce the auxiliary fields $X^a$ as above, whose equations of motion impose the flatness condition.

The $X^a$ can be interpreted as a set of dynamical coordinates, which is apparent from the equations of motion for the vielbein

$$\partial_\alpha X^a = e_\alpha{}^a - \mu\, \epsilon_{\alpha\beta}\epsilon^{ab}T^\beta{}_b\,. \qquad (2.3)$$

If the worldsheet metric is chosen to be Minkowski, $e_\alpha{}^a = \delta_\alpha{}^a$, then the matter dynamics is the same as in the absence of the coupling to 'gravity.' The physical effect of the $T\bar{T}$ deformation is obtained by transforming to a gauge where the worldsheet coordinates $\sigma^\alpha$ are aligned with the target space ones, $X^a$, which are, in fact, the physical coordinates of the $T\bar{T}$-deformed QFT. The above stress-tensor dependent coordinate transformation between worldsheet and target space is ultimately responsible for the $T\bar{T}$ dressing of the S-matrix [10].

This definition of the $T\bar{T}$ deformation, which holds at the full non-linear level, has passed several non-trivial checks. In particular, the partition function obtained by evaluating the path integral satisfies the geometric flow equation (1.1), which is equivalent to the flow equation for the $T\bar{T}$-deformed energy levels [10].

## 2.2 The $JT_a$ deformation from a mixed gauge-vielbein coupling

The $JT_a$ case can be treated analogously. Starting from (1.2), one follows the Hubbard-Stratonovich procedure by coupling the $U(1)$ current to an external gauge field $a_\alpha$ and the stress tensor to an external vielbein. For a treatment of the $J\bar{T}$ case, see [25]. Only the vielbein components parallel to the deformation end up playing a role, so we can restrict the path integral to this "half" of the vielbein. By a careful analysis, along the lines of [12], it is likely possible to prove that the only modes that survive in the path integral are the pure gauge modes of $a_\alpha$ and those of the parallel vielbein. However, the strategy that we have chosen to adopt herein is to simply guess that this will be the case and then show that the QFT so defined satisfies all the consistency requirements expected of it.

Introducing a pair of auxiliary fields to enforce the flatness conditions, we are led to the following proposal:

$$S_{JT_a} = \int d^2\sigma\, e\left[ \mathcal{L}_{\text{QFT}}(\varphi, a_\alpha, e_\alpha{}^a) + \Lambda_a\, \epsilon^{\alpha\beta} (\partial_\alpha X^a - e_\alpha{}^a)(\partial_\beta \Phi - a_\beta) + \Lambda_a^\perp \lambda^\alpha (e_\alpha{}^a - \delta_\alpha{}^a) \right], \tag{2.4}$$

where the vectors $\Lambda_a$ and $\Lambda_a^\perp$ satisfy

$$\Lambda_a \mu^a = 1, \qquad \Lambda_a^\perp \mu^a = 0 \tag{2.5}$$

and $\lambda^\alpha$ are Lagrange multipliers. It will be useful to introduce the unit vectors $n^a$, so that $\mu^a = \mu n^a$, $\Lambda_a = \sigma \mu^{-1} n_a$, where $\sigma = n_a n^a = \pm 1$ depends on whether the deformation is timelike or spacelike. We also define

$$X^\| \equiv \sigma n_a X^a = \mu \Lambda_a X^a, \qquad e_\alpha{}^\| \equiv \sigma e_\alpha{}^a n_a = \mu \Lambda_a e_\alpha{}^a, \tag{2.6}$$

where $\mu$ is the modulus of the vector. The null case can be treated separately.

In analogy with the $T\bar{T}$ deformation, the $\Phi$ and $X^\|$ equations of motion set $a_\alpha$ and $e_\alpha{}^\|$ to have vanishing field strength, $\partial_{[\alpha} a_{\beta]} = \partial_{[\alpha} e_{\beta]}{}^\| = 0$, whereas the $\lambda^\alpha$ Lagrange multiplier sets the components of the vielbein that are perpendicular to $\mu^a$ to be trivial or, more generally, fixed to some pre-specified value. The classical equations of motion for the fields $\Phi$ and $X^a$ arise from varying the action with respect to the background fields $a_\alpha$ and $e_\alpha{}^\|$ [4]

$$\partial_\alpha \Phi = a_\alpha + \mu^a \epsilon_{\alpha\beta} T^\beta{}_a, \qquad \partial_\alpha X^\| = e_\alpha{}^\| + \mu\, \epsilon_{\alpha\beta}\left( J^\beta - \frac{k}{4\pi} \epsilon^{\beta\gamma} a_\gamma \right), \tag{2.7}$$

where $k$ is the coefficient of a potential anomaly for the $U(1)$ symmetry, and the last term arises from the variation of the path integral measure with respect to $a_\alpha$. Note that thanks to the conservation equations for the stress tensor and the current (taking into account the anomaly), these equations are entirely consistent with the flatness of the gauge potentials $a_\alpha$ and $e_\alpha{}^a$.

The interpretation of $X^\|$ and $\Phi$ is again that of dynamical coordinates on physical space and on the $U(1)$ gauge orbit. This allows one to interpolate between the original QFT, where $a_\alpha = 0$ and $e_\alpha{}^a = \delta_\alpha{}^a$, and the $JT_a$-deformed one, where $\Phi = 0$ and the worldsheet coordinates are aligned with the target space ones, $\sigma^\alpha = X^a$. As we discuss in section 4.1, this coordinate transformation is responsible for the S-matrix dressing.

## 3 The partition function

In this section we evaluate the Euclidean path integral over auxiliary fields of the theory defined in (2.4). To do this, we follow the steps set out by [13] for the $T\bar{T}$ deformation, and show

---

[4] Our conventions for the variations in Lorentzian signature are: $\delta S_{\text{QFT}} = \int d^2x\, e\left[ T^\mu{}_a \delta e_\mu{}^a - J^\mu \delta a_\mu \right]$.



that the $JT_a$-deformed partition function satisfies a geometric flow equation. We subsequently match this flow equation to the flow of the energy levels previously obtained in the literature, by rewriting it in terms of the physical parameters that enter the Hilbert space interpretation of the partition function.

## 3.1 The path integral over auxiliary fields

We start with the Euclidean path integral of a $JT_a$-deformed field theory, given by the following expression

$$Z_{JT_a} = \int \frac{\mathcal{D}e^\| \mathcal{D}a \mathcal{D}\Phi \mathcal{D}X^a}{V_{\text{Diff}\|} V_{\text{Gauge}}} e^{\Lambda_a^E \int d^2\sigma \, e \left[ \epsilon^{\alpha\beta}(\partial_\alpha \Phi - a_\alpha)(\partial_\beta X^a - e_\beta{}^a) \right]} Z_0(a_\alpha, e_\alpha{}^\|), \qquad (3.1)$$

where we have already performed the trivial integrals over the Lagrange multipliers $\lambda^\alpha$ and the perpendicular vielbein $e_a{}^\perp$. As explained in section 2, we must be careful to denote by $\Lambda_a^E = (\mu_E^a)^{-1}$ the inverse *Euclidean* $JT_a$ coupling.

Instead of performing the entire path integral over vielbeine and gauge fields, our goal will be simpler: that of deriving the flow equations satisfied by the torus partition function $Z_{JT_a}$ with respect to the external parameters $\mu_E^a$ and the data of the physical target space torus. It will therefore suffice to carefully evaluate the path integral over $X^a$ and $\Phi$, while leaving the remaining functional integrals unevaluated. Our analysis will closely follow [33], and we take our Euclidean worldsheet coordinates to lie in $\sigma^\alpha \in [0, \ell)$.[5]

We decompose the $X^a$ and $\Phi$ in terms of a background field and a fluctuation

$$X^a = \frac{1}{\ell} L_\alpha^a \sigma^\alpha + Y^a(\sigma), \qquad \Phi = \frac{1}{\ell} \nu_\alpha \sigma^\alpha + \phi(\sigma), \qquad (3.2)$$

where the $Y^a(\sigma)$ and $\phi(\sigma)$ are periodic functions on the worldsheet. The background fields $L_\alpha^a$ are maps from the worldsheet onto the target space torus and parametrize the lengths this torus. As we will see, the $\nu_\alpha$ will be interpreted as background chemical potentials for the $U(1)$ current.

In terms of $Y^a$ and $\phi$, the Euclidean action takes the form

$$S_{JT_a} = -\frac{1}{\mu_E} \int d^2\sigma \, e \, \epsilon^{\alpha\beta} \left[ \frac{1}{\ell^2} \left( \nu_\alpha - \ell \, a_\alpha \right) \left( L_\beta^\| - \ell \, e_\beta{}^\| \right) - \partial_\alpha \phi \, e_\beta{}^\| + \partial_\alpha Y^\| \, a_\beta \right], \qquad (3.3)$$

where we remind the reader that for any quantity $B$, we have defined, now in Euclidean signature, $B^\| = n_a B^a = \mu_E \Lambda_a^E B^a$, where $\mu_E$ is the euclidean $JT_a$ coupling. In (3.3) we have dropped all total derivatives of non-winding fields, such as $e \, \epsilon^{\alpha\beta} \, \partial_\alpha \phi \, L_\beta^\|$, whose contribution vanishes. Integrating by parts the last two terms, one obtains a delta function setting to zero the functions that multiply them. Concretely, if we split the field $\phi(\sigma) = \bar{\phi} + \phi'(\sigma)$ into a constant mode $\bar{\phi}$ and the non-zero modes $\phi'$, the path integral over $\mathcal{D}\phi$ evaluates to

$$\int \mathcal{D}\phi \, \exp\left[ -\frac{1}{\mu_E} \int d^2\sigma \, \tilde{\epsilon}^{\alpha\beta} \, \phi' \, \partial_\alpha e_\beta{}^\| \right] = \sqrt{\frac{\tilde{\mathcal{A}}}{2\pi}} \det'(2\pi\mu_E \, \mathbb{I}) \, \delta \left( \epsilon^{\alpha\beta} \partial_\alpha e_\beta{}^\| \right) \int d\bar{\phi}, \qquad (3.4)$$

where det is a functional determinant over a constant $(2\pi\mu_E)$ times an infinite-dimensional identity matrix, $\mathbb{I}$, and the prime denotes the fact that we have excluded the zero mode. The overall normalization factor involving the area of the worldsheet torus, $\tilde{\mathcal{A}} = \int d^2\sigma \, e$, can

---

[5]We collect our conventions here for transparency. We will take $[\sigma^\alpha] = [X^a] = [\mu^a] = [\text{length}]$, $[J^\alpha] = [a_\alpha] = [\Lambda_a] = [\text{length}]^{-1}$, $[e_\alpha{}^a] = [\Phi] = [\nu_\alpha] = [\text{length}]^0$, and as usual $[T_{aa}] = [\text{length}]^{-2}$.

be carefully derived following [33], but will not be needed in our analysis, since it does not depend on the external parameters.

Now we must deal with the formally infinite functional determinant $\det'(2\pi\mu_E\,\mathbb{I})$. The determinant $\det'$ with the zero mode excluded can be trivially related to the determinant over all the modes via:

$$C\,\det{}'(C\,\mathbb{I}) = \det(C\,\mathbb{I}), \tag{3.5}$$

for any constant $C$. As explained in [33], the functional determinant of a pure constant times $\mathbb{I}$ can absorbed into a shift of the (infinite) vacuum energy, for example by a simple modification of the normalization choice for the path integral over $\phi$. One way to see this is to note that when dynamical gravity is turned on, the functional determinant of a constant may be dealt with by the addition of a counterterm of the form

$$S_{\text{ct}} = \int d^2\sigma\, e\, c\,, \tag{3.6}$$

where $c$ is chosen to cancel the divergence. This is simply a renormalization of the world-sheet cosmological constant. In the following, we choose $c$ so as to cancel the infinite factor $\det(2\pi\mu_E\,\mathbb{I})$, and obtain

$$\int \mathcal{D}\phi\, \exp\left[-\frac{1}{\mu_E}\int d^2\sigma\, e\, \epsilon^{\alpha\beta}\,\phi'\,\partial_\alpha e_\beta{}^{\|}\right] = \frac{1}{2\pi\mu_E}\sqrt{\frac{\tilde{\mathcal{A}}}{2\pi}}\,\delta\left(\epsilon^{\alpha\beta}\partial_\alpha e_\beta{}^{\|}\right)\int d\bar{\phi}\,. \tag{3.7}$$

Since the gauge group is compact, $\int d\bar{\phi} = 2\pi$.

The path integral over $Y^a$ splits into a path integral over $Y^{\|}$ and one over $Y^{\perp}$. The steps in performing the path integral over $Y^{\|}$ follow exactly those for $\phi$, and the end result is given by (3.7) with the appropriate replacements, $e_a{}^{\|} \to a_\alpha$ and $\bar{\phi} \to \bar{Y}^{\|}$. The path integral over $Y^{\perp}$ is divergent, having no action functional. Howver, only the zero-modes of this functional integral can contribute something physically relevant (i.e., which depends on the external parameters), since the range of $\bar{Y}^{\perp}$ is fixed by the size of the physical torus. We thus regulate it by dividing out by the non-zero modes:

$$\int \mathcal{D}Y^{\perp} \to \int d\bar{Y}^{\perp}\,. \tag{3.8}$$

Finally, we use the fact that the integral over $Y^a$ zero modes is given by

$$\int d\bar{Y}^{\|}d\bar{Y}^{\perp} = L_\tau^\tau L_\phi^\phi - L_\tau^\phi L_\phi^\tau \equiv \mathcal{A}\,, \tag{3.9}$$

where $\mathcal{A}$ is the area of the target space torus. Putting everything together, we obtain

$$\begin{aligned}
Z_{JT_a} = {} & \frac{\mathcal{A}}{4\pi^2\mu_E^2}\int \frac{\mathcal{D}e^{\|}\mathcal{D}a}{V_{\text{Diff}\|}V_{\text{Gauge}}}\,\tilde{\mathcal{A}}\,e^{\frac{1}{\mu_E}\int d^2\sigma\, e\, \epsilon^{\alpha\beta}\frac{1}{\ell^2}(\nu_\alpha-\ell\,a_\alpha)\left(L_\beta^{\|}-\ell\,e_\beta{}^{\|}\right)} \\
& \times \delta\left(\epsilon^{\alpha\beta}\partial_\alpha e_\beta{}^{\|}\right)\delta\left(\epsilon^{\alpha\beta}\partial_\alpha a_\beta\right)Z_0(a_\alpha, e_\alpha{}^{\|})\,.
\end{aligned} \tag{3.10}$$

## 3.2 The flow equation

We would now like to show that the partition function above obeys a simple diffusive flow equation, similar to the one derived for the case of $T\bar{T}$-deformed QFTs. To do so, we note that the functional integrals in (3.10) localize over constant field configurations $\bar{e}_\alpha{}^{\|}$ and $\bar{a}_\alpha$ of the vielbein and gauge field, as a result of the explicit delta functions. Dropping all the terms that

do not depend on the external parameters, we can schematically write the partition function as

$$Z_{JT_a} \propto \frac{\mathcal{A}}{\mu_E^2} \int \mathcal{D}\bar{e}^{\parallel} \mathcal{D}\bar{a} \; e^{\frac{1}{\mu_E} \tilde{\epsilon}^{\alpha\beta}(\nu_\alpha - \ell\,\bar{a}_\alpha)(L_\beta^{\parallel} - \ell\,\bar{e}_\beta^{\parallel})} Z_0(\bar{a}_\alpha, \bar{e}_\alpha{}^{\parallel}) \,, \tag{3.11}$$

where we have performed the integral over worldsheet coordinates in the exponential and introduced the Levi-Civita symbol $\tilde{\epsilon}^{\alpha\beta} = e\,\epsilon^{\alpha\beta}$, whose components are just numbers. Defining

$$\Psi \equiv \frac{Z_{JT_a}}{\mathcal{A}} \tag{3.12}$$

it is easy to verify that $\Psi$ satisfies the following flow equation:

$$\boxed{\frac{\partial \Psi}{\partial \mu_E} = \tilde{\epsilon}_{\alpha\beta} \frac{\partial^2}{\partial L_\alpha^{\parallel} \partial \nu_\beta} \Psi \,.} \tag{3.13}$$

The above equation can also be written in terms of the individual euclidean time and space components of the coupling vector $\mu_E^a = \left( \mu_E^\tau, \mu_E^\phi \right)$, as

$$\sum_a n^a \left( \partial_{\mu_E^a} - \tilde{\epsilon}_{\alpha\beta} \, \partial_{L_\alpha^a} \partial_{\nu_\beta} \right) \Psi = 0 \,. \tag{3.14}$$

The flow equation (3.13), which we have derived from the path integral definition of the theory, does not imply that each term in the sum must individually vanish. We expect, however, that such a stronger condition holds, since the direction of $\mu^a$ can be chosen arbitrarily. In other words, we have so far defined the $JT_a$ deformation by picking a *fixed* direction $n^a$ in the two-dimensional space of possible $JT_a$ couplings, and flowed a specified amount $\mu_E$ along it. Because the $JT_a$ deformations with respect to different directions commute [20], we could have alternatively reached the same final deformed QFT by following along *any* curve in the space of couplings that has the same endpoints. This freedom in choosing the path in coupling space would be reflected in the definition (1.2) of the $JT_a$ deformation by requiring that it hold not only for a fixed direction of the vector $\mu^a$, but rather, independently for each of its components. This likely translates into two independent flow equations for the partition function, which correspond to the vanishing of each individual term in (3.14).[6] It is precisely these individual equations that we will match as a flow on the energy spectrum.

*Hilbert space interpretation*

To gain a physical understanding of the above flow equation and to match with the flow equations for the energy spectrum that were previously derived in the literature [20] , we pass to the Hilbert space interpretation of the torus partition function:

$$Z_{JT_a} = \text{Tr}\left[ e^{-\beta E + i\theta P + i\eta Q} \right], \tag{3.15}$$

where

$$E = -\int_0^R d\phi \, \langle T_{\tau\tau} \rangle \,, \quad P = i\int_0^R d\phi \, \langle T_{\tau\phi} \rangle \,, \quad Q = i\int_0^R d\phi \, \langle J_\tau \rangle \,, \tag{3.16}$$

---

[6]This suggests that the path integral proposed by [32] corresponds to choosing a contour in coupling space where first one coupling is increased from zero to a finite value, then the orthogonal coupling is turned on. It is clear, at least heuristically, that this procedure will produce both orthogonal components of the vielbein, with a measure that will coincidentally appear Lorentz-invariant, and two independent gauge fields, one for each leg of the integration contour.

are the energy, momentum, and the conserved $U(1)$ charge of the states we sum over, while $(\beta, \theta, \eta)$ are the chemical potentials that couple to them. The momentum is quantized in units of $1/R$, where $R$ is the size of the spatial circle of the torus, whereas $Q$ is assumed to be integer quantized[7]

$$P = \frac{2\pi p}{R}, \quad Q, p \in \mathbb{Z}. \tag{3.17}$$

Since the $JT_a$ deformation explicitly breaks Lorentz invariance, the conserved stress tensor is not symmetric along the flow. In order to keep track of its asymmetry, [19] have proposed to introduce a coupling to an explicit background vielbein $v_E$ that the energy levels will depend on, so that

$$\langle T_{\phi\tau} \rangle = \frac{1}{R} \frac{\partial E}{\partial v_E}, \tag{3.18}$$

where we took into account the fact that the expectation value of the stress tensor on the torus is independent of $\phi$ and $v_E$ is an Euclidean vielbein. In a similar vein, we introduce a background gauge field $a^\phi$ that couples to the operator $J_\phi$ in the Hamiltonian, so that

$$\langle J_\phi \rangle = \frac{1}{R} \frac{\partial E}{\partial a^\phi}. \tag{3.19}$$

Thus, the dependence of the partition function on the deformation parameter(s) and the background fields not appearing explicitly in (3.15) is encoded in the dependence of the energy on them, namely

$$E = E\left(\mu_E^a, R, v_E, a^\phi\right). \tag{3.20}$$

As explained above, we assume that $Z_{JT_a}$ depends independently on $\mu_E^\tau, \mu_E^\phi$. As usual, we have

$$\langle T_{\phi\phi} \rangle = -\frac{\partial E}{\partial R}. \tag{3.21}$$

We would now like to recast the flow equation (3.13) for the partition function in terms of the "physical" parameters appearing in the Hilbert space definition (3.15).

The relationship between the parameters $L_\alpha^a$ and $v_\alpha$ appearing in our flow equation, and the physical parameters $(\beta, \theta, \eta, R, v_E, a^\phi)$ has a very simple geometrical origin. Recall that the coordinates $X^a$ are maps from the woldsheet into the target space, classically given by

$$X^a = \frac{1}{\ell} L_\alpha^a \sigma^\alpha. \tag{3.22}$$

If, for the moment, we turn off the worldsheet vielbein $v_E$, then we have the usual identification

$$L_\tau^{\hat{a}} = \begin{pmatrix} \beta \\ \theta \end{pmatrix}, \qquad L_\phi^{\hat{a}} = \begin{pmatrix} 0 \\ R \end{pmatrix}, \tag{3.23}$$

where the index $\hat{a}$ indicates that we are temporarily working with a target space metric of the form $g_{\hat{a}\hat{b}} = \delta_{\hat{a}\hat{b}}$ and we have aligned the worldsheet $\phi$ direction with the target space one.[8] Turning on $v_E$ means that we now consider the $JT_a$-deformed QFT on a background metric that is not diagonal. This corresponds to going from a vielbein $e^a_{\hat{a}} = \delta^a_{\hat{a}}$ to

$$e^a_{\hat{a}} = \begin{pmatrix} 1 & v_E \\ 0 & 1 \end{pmatrix} \quad \Rightarrow \quad L_\alpha^a \equiv e^a_{\hat{a}} L_\alpha^{\hat{a}} = \begin{pmatrix} \beta + v_E\theta & v_E R \\ \theta & R \end{pmatrix}. \tag{3.24}$$

Notice that $\mathcal{A} = \det L_\alpha^a = \beta R$.

---

[7]Note that in the case of the $J\bar{T}$ deformation, the current whose charge is conserved will not be chiral [27].

[8]It is not clear whether we can consider other options in a Lorentz-breaking theory.

The background chemical potentials $\nu_\alpha$ are defined using $\Phi = \ell^{-1} \nu_\alpha \sigma^\alpha = \ell^{-1} a_a X^a$, where $a_a$ are the components of a background gauge field on the physical torus. Using (3.22), we find

$$\nu_\alpha = a_a L_\alpha^a = a_{\hat{a}} L_\alpha^{\hat{a}}. \tag{3.25}$$

The parameter $\eta$ appearing in the partition function is, by definition, equated with $\beta a_{\hat{\tau}}$ in the frame where the physical metric is $\delta_{\hat{a}\hat{b}}$, so we have

$$\nu_\tau = \beta a_{\hat{\tau}} + \theta\, a_{\hat{\phi}} = \eta + \theta\, a_{\hat{\phi}}\,, \qquad \nu_\phi = R a_{\hat{\phi}}\,. \tag{3.26}$$

The relationship between $a_{\hat{\phi}}$ and the background field $a^\phi$ is given by

$$a^a = e^a{}_{\hat{a}}\, a^{\hat{a}} \quad \Rightarrow \quad a^\phi = a^{\hat{\phi}} = a_{\hat{\phi}}. \tag{3.27}$$

The equations (3.24) and (3.26) specify the relationship between $L_\alpha^a$, $\nu_\alpha$ and the physical parameters. The inverse relations read

$$R = L_\phi^\phi\,, \quad \theta = L_\tau^\phi\,, \quad \beta = \frac{L_\tau^\tau L_\phi^\phi - L_\tau^\phi L_\phi^\tau}{L_\phi^\phi}\,, \quad \nu_E = \frac{L_\phi^\tau}{L_\phi^\phi}\,, \quad a^\phi = \frac{\nu_\phi}{L_\phi^\phi}\,, \quad \eta = \frac{\nu_\tau L_\phi^\phi - \nu_\phi L_\tau^\phi}{L_\phi^\phi}\,. \tag{3.28}$$

In terms of these variables, the flow equations for the reduced partition function $\Psi \equiv Z_{JT_a}/\mathcal{A}$ may be written as

$$\frac{\partial \Psi}{\partial \mu_E^\tau} = \left(\partial_{L_\tau^\tau} \partial_{\nu_\phi} - \partial_{L_\phi^\tau} \partial_{\nu_\tau}\right) \Psi = \frac{1}{R}\left(\partial_\beta \partial_{a^\phi} - \partial_\eta \partial_{\nu_E}\right)\Psi\,, \tag{3.29}$$

$$\frac{\partial \Psi}{\partial \mu_E^\phi} = \left(\partial_{L_\tau^\phi} \partial_{\nu_\phi} - \partial_{L_\phi^\phi} \partial_{\nu_\tau}\right)\Psi = \left[\frac{\nu}{R}\left(\partial_\eta \partial_{\nu_E} - \partial_\beta \partial_{a^\phi}\right) - \partial_R \partial_\eta - \frac{1}{R}\left(\theta\, \partial_\eta \partial_\theta - \partial_{a^\phi} \partial_\theta + \partial_\eta\right)\right]\Psi\,. $$

The above can also be written as a set of flow equations for the individual energy levels, $E_n$, using (3.15) and the fact that the momentum $P_n$ is quantized in units of $1/R$

$$\frac{\partial E_n}{\partial \mu_E^\tau} = \frac{1}{R}\left[-iQ_n \frac{\partial E_n}{\partial \nu_E} - E_n \frac{\partial E_n}{\partial a^\phi}\right]\,, \tag{3.30}$$

$$\frac{\partial E_n}{\partial \mu_E^\phi} = -iQ_n\left(\frac{\partial E_n}{\partial R} - \frac{\nu_E}{R}\frac{\partial E_n}{\partial \nu_E}\right) + \frac{iP_n + \nu_E E_n}{R}\frac{\partial E_n}{\partial a^\phi}\,. \tag{3.31}$$

Upon an appropriate mapping of the parameters, these are precisely[9] the equations given in (3.18) of [20]. Notice that, despite appearances, the above flow equations are real, given that $\mu_E^\phi$ and $\nu_E$ are purely imaginary. Letting $\mu_E^\phi = -i\mu^\phi$, $\mu_E^\tau = \mu^t$ and $\nu_E = i\nu$, the manifestly reality of the flow equations can be seen from:

$$\frac{\partial E_n}{\partial \mu^t} = -\frac{1}{R}\left[Q_n \frac{\partial E_n}{\partial \nu} + E_n \frac{\partial E_n}{\partial a^\phi}\right]\,, \qquad \frac{\partial E_n}{\partial \mu^\phi} = -Q_n\left(\frac{\partial E_n}{\partial R} - \frac{\nu}{R}\frac{\partial E_n}{\partial \nu}\right) + \frac{P_n + \nu E_n}{R}\frac{\partial E_n}{\partial a^\phi}\,. \tag{3.33}$$

Notice the above flow equations should hold in an arbitrary QFT, since they follow from the flow equations for the partition function, whose definition is insensitive to whether the seed theory is a CFT or a generic QFT, even a non-relativistic one.

---

[9]Note that since [20] take derivatives with the combination $\tilde{\mathbb{P}}^1 \equiv -Ra^\phi$ held fixed, we must be careful to also replace:

$$\partial_R^{\mathrm{fr}} \to \partial_R - \frac{a^\phi}{R}\partial_{a^\phi}\,, \qquad \partial_{\mathbb{P}^1} \to -\frac{1}{R}\partial_{a^\phi}\,. \tag{3.32}$$

### 3.3 Explicit solution to the flow equations

Let us now review, following [20], how one solves the above flow equations to obtain a solution for the deformed energies as a function of the undeformed ones and the other conserved charges. These solutions have been previously found in the literature, but since most authors concentrated on having many different deformations turned on at the same time, they are rather cumbersome to write down. Thus, in this section we solve the flow equations for the case of the pure $JT_a$ deformation, for which we will see the solution takes a very simple form.

The first step is to notice that, as was the case for $T\bar{T}$, the general solution to equations (3.30-3.31) is given simply by shifting the parameters according to

$$E_n(\mu^a, R, v, a^\phi) = E_n^{(0)}\left(R - \mu^\phi Q_n, \frac{vR - \mu^t Q_n}{R - \mu^\phi Q_n}, a^\phi + \frac{\mu^\phi(P_n + vE_n) - \mu^t E_n}{R - \mu^\phi Q_n}\right), \tag{3.34}$$

where from now on we work in terms of the Lorentzian parameters, which are manifestly real. Notice that if we write the solution not in terms of $R, v, a^\phi$ as in the previous literature, but in terms of $R, Rv$ and $a^\phi R$ which, as we saw, have a more invariant interpretation, the above shifts simply correspond to

$$\boxed{R \to R - \mu^\phi Q, \quad vR \to vR - \mu^t Q, \quad Ra^\phi \to Ra^\phi - \mu^t E + \mu^\phi P + \mu^\phi vE.} \tag{3.35}$$

In section 4, we will see how these simple shifts are reproduced from the TBA equations.

Thus, if we know $E_n^{(0)}$ as a function of general background parameters, (3.34) provides us with an algebraic equation for the deformed energy levels, $E_n$. These expressions have been worked out explicitly in e.g. [19] for the special case of a seed CFT. A simple way to understand it is to notice that in absence of a vielbein but in presence of an external gauge potential $a_{\hat{\phi}}$, the expression for the energy levels in a CFT on a circle of circumference $R$ is given by

$$E_n^{(0)} = \frac{2\pi\Delta_n}{R} + \frac{kR}{16\pi}a_{\hat{\phi}}^2, \tag{3.36}$$

where $\Delta_n \in \mathbb{R}$ is the associated conformal dimension in the CFT on the plane and $k$ is the coefficient of the chiral anomaly. The second term represents the shift in the energy levels due to the external gauge potential. Once we turn on a background vielbein $v$, the expression becomes [19]:

$$E_n^{(0)}(R, v, a^\phi) = \frac{1}{1 - v^2}\left(\frac{2\pi\Delta_n}{R} + \frac{2\pi p_n}{R}v - Q_n v a^\phi + \frac{kR}{16\pi}(a^\phi)^2\right), \tag{3.37}$$

where we have plugged in the expression (3.17) for $P_n$ and the third term can be understood as a shift in $a^\tau$ proportional to $va_{\hat{\phi}} = va^\phi$. We will be interested in the spectrum of the deformed QFT with all background fields switched off, so $v = a^\phi = 0$ in (3.34) and $\mu^t = -\mu_t, \mu^\phi = \mu_\phi$. Plugging (3.37) into (3.34) yields the following quadratic equation for the energy levels:[10]

$$\frac{k\mu_t^2}{8\pi}E_n^2 - 2\left(R - \mu_\phi Q_n - \mu_t\mu_\phi\frac{kP_n}{8\pi}\right)E_n + 2E_n^{(0)}R + \frac{k\mu_\phi^2}{8\pi}P_n^2 + 2\mu_t P_n Q_n = 0, \tag{3.38}$$

---

[10]More generally, $\tilde{k}\gamma^2 E_n^2 - \left[\gamma(Q_n v + 2\tilde{k}d) + \bar{R}(1 - v^2)\right]E_n + E_n^{(0)}R + b(Rv - \mu^t Q_n) + \tilde{k}d^2 = 0$, where

$$\tilde{k} \equiv \frac{k}{16\pi}, \qquad \gamma \equiv \mu^t - v\mu^\phi, \qquad \bar{R} \equiv R - \mu^\phi Q_n, \qquad d \equiv a^\phi\bar{R} + \mu^\phi P_n, \qquad b \equiv P_n - a^\phi Q_n.$$

with solution

$$E_n = -\frac{\mu_\phi}{\mu_t} P_n + \frac{8\pi}{k\mu_t^2}\left[ R - \mu_\phi Q_n - \sqrt{(R - \mu_\phi Q_n)^2 - \frac{k\mu_t}{4\pi}\left(R\left(\mu_t E_n^{(0)} + \mu_\phi P_n\right) - P_n Q_n(\mu_\phi^2 - \mu_t^2)\right)} \right],$$
(3.39)

where the sign in front of the square root is fixed by the requirement that the spectrum reduces to the undeformed one as $\mu_{t,\phi} \to 0$. The spectrum of the $J\bar{T}$ or $\bar{J}T$ deformations is then obtained by taking $\mu_t = \mp\mu^\phi = \mu$. Notice also that in the case $\mu_t = 0$, the quadratic equation reduces to a linear one.

It is interesting to note that the spectrum's dependence on the anomaly coefficient $k$ descendes entirely from the $k$ dependence of the CFT energy (3.37) in presence of the background fields, as the flow equation (3.13) does not contain any explicit factor of $k$. This is quite different from the flow equation for $J\bar{T}$-deformed CFTs obtained by [28], which depends on $k$ in a rather complicated fashion. One possible reason for the different form of the flow equation we find is that while in [28] the partition sum was over the charge associated with the strictly chiral current, which satisfies a non-trivial flow equation itself, our partition function sums instead over an integer-quantized charge, which is associated with a possibly non-chiral current. One way to ensure that the current is exactly chiral is to consider, as in [20], a joint $JT_a$ and $\tilde{J}T_a$ deformation with equal coefficients, where $\tilde{J} = \star J$ is the dual current to $J$. In this case, the flow equations for the energy start depending explicitly on the anomaly coefficient [19, 20], and likely the same is true of the flow equations for the partition function.

# 4 S-matrix dressing factor and the TBA

In this section, we consider the S-matrix of $JT_a$-deformed QFTs. In the case of the $T\bar{T}$ deformation, it is now well understood [5,6,10] that the effect of the dynamical change of coordinates (2.3) is to simply dress the S-matrix of the undeformed QFT by a particular Castillejo-Dalitz-Dyson (CDD) phase.

Now that we have shown that the $JT_a$ deformation can be understood as coupling the original QFT to a dynamical vielbein and a gauge field, we expect that the $JT_a$ deformation should have a similar effect on the S-matrix of the undeformed QFT via a particular, and universal, CDD dressing. In this section, we derive the dressing factor that follows from our definition (2.4), closely following [10]. We then show, following [8], that the shifts (3.35) in the spectra can be easily recovered by applying the Thermodynamic Bethe Ansatz (TBA) equations [34] to the dressed S-matrix.

## 4.1 Derivation of the dressing factor

In order to see the effect of the $JT_a$ deformation on the S-matrix, we need a notion of asymptotic states. In the asymptotic past, we can decompose our QFT fields $\psi$ in free-field modes

$$\psi_{in} = \int_{-\infty}^{\infty} \frac{dp}{\sqrt{4\pi\omega_p}}\left(a_{in}^\dagger(p)e^{ip_\alpha\sigma^\alpha + iq\int a} + a_{in}(p)e^{-ip_\alpha\sigma^\alpha - iq\int a}\right)\Bigg|_{p^0 = \omega_p},$$
(4.1)

where the subscript "in" indicates that the creation-annihilation operators above act on the in-state vacuum and we have included a background pure gauge field $a_\alpha$.

The effect of deforming the original QFT by the $JT_a$ operator is captured by the introduction of a new set of dynamical coordinates $X^a$, with respect to which the scattering is measured, and of a stress-tensor dependent gauge transformation, given in (2.7). As discussed, there

only is one nontrivial dynamical coordinate in the direction parallel to $\mu^a$, denoted $X^{||}$ and defined in (2.6). In the $JT_a$ description, the same free field operator $\psi$ can be written as

$$\psi_{in} = \int_{-\infty}^{\infty} \frac{dp}{\sqrt{4\pi\omega_p}} \left(A_{\text{in}}^\dagger(p)e^{ip_aX^a+iq\Phi} + A_{\text{in}}(p)e^{-ip_aX^a-iq\Phi}\right)\bigg|_{p^0=\omega_p}, \qquad (4.2)$$

where $A_{in}^\dagger$ are creation operators that define the in-state in the $JT_a$ description. Setting the vielbein $e_\alpha{}^a = \delta_\alpha{}^a$ in (2.7) and letting $X^{||} = \sigma^{||} + Y^{||}(\sigma)$, we find the following relation

$$A_{\text{in}}^\dagger(p,q) = a_{\text{in}}^\dagger e^{-ip_{||}Y^{||}-iq\Delta\Phi}, \qquad (4.3)$$

where the additional phases come from the fact that we are now measuring scattering with respect to the dynamical coordinates $X^a$, and the background gauge potentials also differ by the amount given in (2.7). The equations for $Y^{||}$ and $\Delta\Phi$ are

$$\partial_\alpha Y^{||} = \mu \, \epsilon_{\alpha\beta} J^\beta, \qquad \partial_\alpha \Delta\Phi = \mu \, \epsilon_{\alpha\beta} T^\beta{}_{||}. \qquad (4.4)$$

Labeling $\sigma^\alpha = (t,x)$ and choosing $\epsilon_{tx} = 1$, the solution for $Y^{||}$ and $\Delta\Phi$ in the asymptotic past, $t \to -\infty$, is given by

$$\begin{aligned}
Y^{||}(t \to -\infty, x) &= \text{const.} + \mu \int_{-\infty}^{x} dx' J_t(x'), \\
\Delta\Phi(t \to -\infty, x) &= \text{const.} + \mu \int_{-\infty}^{x} dx' T_{t\,||}(x').
\end{aligned} \qquad (4.5)$$

Up to the constant, the above formula for $Y^{||}$ measures the amount of $U(1)$ charge to the left of the point $x$. We may pick this constant in a parity symmetric way such that

$$Y^{||}(x \to \pm\infty) = \pm\frac{\mu Q^{\text{tot}}}{2}, \qquad Q^{\text{tot}} \equiv \int_{-\infty}^{\infty} dx \, J_t, \qquad (4.6)$$

meaning const. $= -\mu Q^{\text{tot}}/2$. Similarly, the formula for $\Delta\Phi$ in the asymptotic past measures, up to the constant, the amount of parallel momentum (to $\mu^a$) that is to the left of the point $x$. Again, we choose the constant such $\Phi(x \to \pm\infty) = \pm\mu P_{||}^{\text{tot}}/2$. To recap, the classical solution of $Y^{||}(t \to -\infty, x)$ and $\Phi(t \to -\infty, x)$ are constructed such that:

$$\begin{aligned}
Y^{||}(t \to -\infty, x) &= \frac{\mu}{2} \left\{(\text{Total charge to left of } x) - (\text{Total charge to right of } x)\right\}, \\
\Phi(t \to -\infty, x) &= \frac{\mu}{2} \left\{\left(\text{Total } P_{||} \text{ to left of } x\right) - \left(\text{Total } P_{||} \text{ to right of } x\right)\right\}.
\end{aligned} \qquad (4.7)$$

Now, if we prepare an in-state using these $A_{\text{in}}^\dagger$ operators, and order the particles with momenta $p^i$ and charges $q^i$ by their rapidities, which corresponds to spatial ordering in the asymptotic past, it is not too difficult to convince oneself that each insertion of $A_{\text{in}}^\dagger(p^k, q^k)$ contributes to the the overall dressing of the in-state $|\{p^i, q^i\}, \text{in}\rangle_{\text{undressed}}$ a phase[11]:

$$e^{-i\frac{\mu}{2}p_{||}^k\left(\sum_{j<k}q^j - \sum_{j>k}q^j\right) + i\frac{\mu}{2}q^k\left(\sum_{j<k}p_{||}^j - \sum_{j>k}p_{||}^j\right)}, \qquad (4.8)$$

where $\mu p_{||} = \mu^a p_a$. The cumulative effect of all these phases is to dress the total state as:

$$|\{p^i, q^i\}, \text{in}\rangle_{\text{dressed}} = \prod_i^n A_{\text{in}}^\dagger(p^i, q^i)|0\rangle = e^{-i\mu \sum_{i<j}\left(q^i p_{||}^j - q^j p_{||}^i\right)}|\{p^i, q^i\}, \text{in}\rangle. \qquad (4.9)$$

---

[11]The reason for the extra relative minus sign between the two factors in the exponent is that $P_{||} = -\sum_i p_{||}^i$, as can be easily checked by taking $\mu^a$ to point in the time direction and requiring that both sides of the equation be positive.

The argument for the out-states is exactly the same, except that they are anti-ordered according to their rapidities, and the phase swaps sign because we use annihilation operators rather than creation operators on the out-state. These two signs cancel. Denoting the dressed S-matrix as $S_{\mu^a}$, the incoming charges and momenta as $q^i, p_a^i$ and the outgoing charges and momenta as $\bar{q}^j, \bar{p}_a^j$, the final result for the dressed S-matrix is

$$S_{\mu^a}\left(\{p^i, q^i\} \to \{\bar{p}^j, \bar{q}^j\}\right) = e^{-i\mu^a \sum_{i<j}\left(q^i p_a^j - q^j p_a^i\right)} e^{-i\mu^a \sum_{i<j}\left(\bar{q}^i \bar{p}_a^j - \bar{q}^j \bar{p}_a^i\right)} S_0\left(\{p^i, q^i\} \to \{\bar{p}^j, \bar{q}^j\}\right).$$
(4.10)

Notice that the above dressing is identical to that found in planar diagrams of noncommutative dipole field theories [2, 3] with $\mu^a$ identified with the dipole length.

## 4.2 Match with the finite-size spectrum via TBA

It is well known that in integrable theories, where scattering is elastic and the $2 \to 2$ S-matrix elements take the form of a phase shift $e^{i\delta(\beta_i, \beta_j)}$, where $\beta_i$ are the particle rapidities, the scattering phase can be related to the finite-size energy spectrum of the theory via the so-called Thermodynamic Bethe Ansatz (TBA) equations [34].

The derivation of these equations proceeds in two steps. First, one considers the partition function of the original Lorentzian QFT on a circle of circumference $R$ and at a temperature $T = L^{-1}$. This partition function can also be evaluated via an euclidean path integral on a torus whose sides have lengths $L, R$. The same path integral can be alternatively interpreted as the partition function of an - in principle different - Lorentzian theory, called the "mirror" theory, placed on a circle of size $L$ and at temperature $R^{-1}$. Since they share the same euclidean path integral representation, the original and the mirror theory are related. We will label quantities such as the free energy or Hamiltonian of the mirror theory with tildes, to distinguish them from quantities in the original picture.

If the original QFT is relativistic, as considered in [34], then the mirror theory is identical to the original one. If, however, the original QFT is not Lorentz invariant, as will be the case for the $JT_a$ - deformed QFTs, then the mirror theory is obtained via a double Wick rotation of the original one. As nicely explained in e.g. [35], this double Wick rotation will in general affect the dispersion relation $E(p)$ of the asymptotic one-particle states, and the mirror one can be obtained by the simple replacement $H \to i\tilde{p}$, $p \to i\tilde{H}$ in the dispersion relation of the original QFT.

Now consider the limit $L \to \infty$ with $L \gg R$. This corresponds to the zero temperature limit of the original theory in which the partition function is dominated by the finite-size ground state energy, $E_0(R)$. In the mirror picture, $L \to \infty$ represents a thermodynamic limit where the system size becomes infinite and the partition function is well approximated by the free energy density at temperature $R^{-1}$, $\tilde{f}(R)$. Thus,

$$\lim_{L \to \infty} Z(R, L) \approx e^{-E(R)L} \approx e^{-R\tilde{f}(R)L} ,$$
(4.11)

so the finite-volume ground state energy of the original QFT equals $R$ times the free energy density in the mirror picture.

Since the mirror theory lives in approximately infinite volume, one has a well-defined notion of asymptotic states. The second step of the derivation consists in doing statistics over a large number of (mirror) particles that scatter, and find the minimum of their free energy, subject to the constraint that the momentum of each particle, $i$, satisfies a quantization condition of the form

$$\tilde{p}_i L + \sum_{j \neq i} \delta(\tilde{p}_i, \tilde{p}_j) = 2\pi n_i , \qquad n_i \in \mathbb{Z},$$
(4.12)

where $L$ is the size of the spatial circle in the mirror theory. Assuming for simplicity that all particles have the same mass $m$ and satisfy a relativistic dispersion relation, the energy and momentum of each particle is given by $\tilde{E} = m \cosh \beta$ and respectively $\tilde{p} = m \sinh \beta$. When a large number of particles is present, it is useful to introduce the density of momentum levels per unit rapidity, $\rho_l(\beta) = dn/d\beta$ and a particle density per unit rapidity, $\rho_p(\beta)$. Upon taking a derivative with respect to $\beta$, the quantization condition (4.12) becomes

$$2\pi \rho_l(\beta) = mL \cosh \beta + \int d\beta' \rho_p(\beta') \frac{\partial \delta(\beta, \beta')}{\partial \beta} \tag{4.13}$$

and the total (mirror) energy, momentum, and charge of the state take the form

$$\tilde{H} = m \int d\beta \, \rho_p(\beta) \cosh \beta \,, \quad \tilde{P} = m \int d\beta \, \rho_p(\beta) \sinh \beta \,, \quad \tilde{Q} = e \int d\beta \, \rho_p(\beta) \,, \tag{4.14}$$

where $e$ is the fundamental unit of charge. The free energy $\tilde{f}(R)$ is obtained by minimizing

$$L\tilde{f}(\rho_l, \rho_p) \equiv \tilde{H}(\rho_p) - \frac{i\tilde{\nu}}{R} \tilde{Q}(\rho_p) - \frac{i\tilde{\theta}}{R} \tilde{P}(\rho_p) - \frac{1}{R} \tilde{S}(\rho_l, \rho_p), \tag{4.15}$$

with respect to the densities $\rho_p$ and $\rho_l$, subject to the constraint (4.13), where we work in an ensemble with chemical potential $\tilde{\nu}$ for the charge $\tilde{Q}$ and fugacity $\tilde{\theta}$ for the momentum. The quantity $\tilde{S}(\rho_l, \rho_p)$ is an entropy which counts the number of ways of distributing $n_p = \rho_p(\beta)\Delta\beta$ particles amongst $N_l = \rho_l(\beta)\Delta\beta$ levels. Considering for concreteness bosonic statistics, we have

$$\tilde{S}(\rho_l, \rho_p) = \int d\beta \left[ (\rho_l + \rho_p) \log(\rho_l + \rho_p) - \rho_l \log \rho_l - \rho_p \log \rho_p \right] . \tag{4.16}$$

Extremizing the free energy with respect to $\rho_{p,l}$ yields the TBA equations for the "pseudoenergy" $\varepsilon(\beta)$

$$\varepsilon(\beta) = Rm \cosh \beta - ie\tilde{\nu} - i\tilde{\theta}m \sinh \beta + \frac{1}{2\pi} \int d\beta' \frac{\partial \delta(\beta', \beta)}{\partial \beta} \log\left(1 - e^{-\varepsilon(\beta')}\right), \tag{4.17}$$

which is defined via the ratio

$$\frac{\rho_p}{\rho_l} \equiv \frac{1}{e^\varepsilon - 1} \, . \tag{4.18}$$

The free energy at the minimum turns out to only depend on this ratio, and reads

$$R\tilde{f}(R) = \frac{m}{2\pi} \int d\beta \, \cosh \beta \log\left(1 - e^{-\varepsilon(\beta)}\right). \tag{4.19}$$

Using (4.11), this equals the vacuum energy $E_0(R)$ of the original theory.[12] The charge in the original model is, similarly, simply given by [30]

$$Q = \frac{e}{2\pi} \int d\beta \log(1 - e^{-\varepsilon(\beta)}), \tag{4.20}$$

where we chose an appropriate normalization.

Let us now consider the specific case of the $JT_a$ deformation, where the phase shift is given by (4.10)

$$\delta_{JT_a}(\beta, \beta') = me\mu_t(\cosh \beta - \cosh \beta') + me\mu^\phi(\sinh \beta - \sinh \beta') . \tag{4.21}$$

---

[12]While we have omitted it in the expression, it should be clear from (4.17) that $\varepsilon$ depends on $R$.

Notice $\delta_{JT_a}(\beta,\beta')$ is antisymmetric. Assuming the original S-matrix of the integrable QFT we deform is given by $e^{i\delta(\beta,\beta')}$, the $JT_a$ deformation induces an additional phase shift $\Delta\delta(\beta,\beta') = \delta_{JT_a}(\beta,\beta')$ into the TBA equation (4.17). Plugging in, we find that the deformed pseudoenergy obeys

$$
\begin{aligned}
\varepsilon(\beta) \;=\; & Rm\cosh\beta - i\tilde{\theta}m\sinh\beta - ie\tilde{\nu} - \frac{em}{2\pi}(\mu_\phi\cosh\beta + \mu_t\sinh\beta)\int d\beta'\ln(1-e^{-\varepsilon(\beta')}) + \\
& + \frac{1}{2\pi}\int d\beta'\frac{\partial\delta(\beta',\beta)}{\partial\beta}\log\left(1-e^{-\varepsilon(\beta')}\right).
\end{aligned}
\tag{4.22}
$$

We immediately notice that the effect of the couplings $\mu_\phi,\mu_t$ is to shift

$$
R \to R - \mu_\phi Q\,, \qquad \tilde{\theta} \to \tilde{\theta} - i\mu_t Q
\tag{4.23}
$$

in the equation for the ground state energy. Noting that $\tilde{\theta}$ in the mirror picture simply corresponds to $-\nu_E R = -i\nu R$ in the original one, this nicely reproduces the first two shifts in (3.35). We used the fact that the mirror theory lives on a Minkowski background, so $\mu^\phi = \mu_\phi$ and $\mu^t = -\mu_t$.

The remaining shift in $Ra^\phi$ can be understood directly in the original theory, as being due to a modification in the quantization condition for the particle momenta. As we already discussed, the phase shift $\delta(\beta,\beta')$ enters the TBA equations via its derivative with respect to $\beta$ and modifies the spectrum. However, the phase shift (4.21) also has a piece $\delta_{ct}$ that is independent of the particle's rapidity $\beta$, which enters the quantization condition for the momenta in the original QFT (compactified on a circle of size $R$) as

$$
p_i R + \delta_{ct} + \sum_{j\neq i}\delta^{nc}(p_i,p_j) = 2\pi n_i,
\tag{4.24}
$$

where $\delta^{nc}$ only contains the explicitly $p_i$ momentum-dependent pieces in the phase shift and $\delta_{ct}$ is obtained by summing over all the other particles that the $i^{th}$ particle interacts with

$$
\delta_{ct} = -e\mu_t E - e\mu_\phi P,
\tag{4.25}
$$

where $E,P$ are the total energy and momentum in the original theory.

The shift by $\delta_{ct}$ can be interpreted as turning on a spatial gauge field $-ea_\phi R = \delta_{ct}$. Setting $\nu = 0$ for simplicity and plugging in (4.25), one obtains that $Ra_\phi \to Ra_\phi + \mu_t E + \mu_\phi P$. This is the same as the shift in (3.35).

We can however go further and understand how this shift affects the TBA. As explained at the beginning of this section, the dispersion relation in the mirror theory is obtained via a double Wick rotation of the dispersion relation in the original QFT. Thus, if we start with the dispersion relation for a free relativistic particle

$$
H = \sqrt{p^2 + m^2},
\tag{4.26}
$$

then turning on a constant spatial gauge field as above modifies the dispersion relation to

$$
H = \sqrt{(p - ea_\phi)^2 + m^2}.
\tag{4.27}
$$

Going to the mirror picture, we send $H \to i\tilde{p}$ and $p \to i\tilde{H}$, which results in the mirror dispersion relation

$$
\tilde{H} = -iea_\phi + \sqrt{\tilde{p}^2 + m^2}.
\tag{4.28}
$$

This shift in each mirror particle's zero point energy is exactly equivalent to introducing a chemical potential $\tilde{\nu} = a_\phi R$ for the mirror charge $\tilde{Q}$, as in (4.15). The interpretation of this

chemical potential is as a temporal Wilson line in the mirror theory, which maps back to a spatial Wilson line in the original one.

Notice that in the analysis above we have only exhibited the relation between the TBA and the shifts (3.35) for the ground state energy. However, as explained e.g. in [8], the excited levels can be easily captured using the same method.

# 5   Final remarks

In this article, we have proposed a path integral definition of general $JT_a$-deformed QFTs by coupling the original QFT to a flat dynamical gauge potential and "half" of a flat dynamical vielbein. We showed that the partition function of the theory so defined satisfies a very simple geometric flow equation, which precisely reproduces the flow equations for the $JT_a$-deformed energy levels that were previously derived in the literature. Also, our definition leads to a universal dressing of the original S-matrix by a phase factor that depends on the charges and momenta of the external particles. For the case of integrable theories, we showed that this dressing factor precisely matches the $JT_a$-induced modification of the finite size spectrum via the TBA equations.

It would be very interesting if this formulation of the theory could be used to study correlation functions. In particular, given that the dressing phase of the S-matrix precisely coincides with that appearing in large $N$ dipole theories [2, 3], it is important to understand whether there exists a concrete relation between $JT_a$-deformed QFTs and dipole-deformed QFTs. It would also be very fruitful to understand the relation between the simple phase factors that appear in the S-matrix and the more complicated structure of the correlation functions in $J\bar{T}$-deformed CFTs [29].

There exist various generalizations of the $JT_a$-deformed QFTs we considered that deserve further study. In particular, we would like to understand the flow equations for the partition function in the case where both $J$ and its dual conserved current $\star J$ are turned on. In this case, the flow equations for the energy presented in [20] indicate an explicit dependence on the anomaly coefficient $k$, which may translate into an explicit dependence of the flow equations for the partition function on it. It would be interesting to see whether one can provide a path integral realization of these models, and whether there is still an intuitive, geometric flow equation that the partition function satisfies.

# Acknowledgments

The authors would like to thank Benjamin Basso, Sergei Dubovsky, Victor Gorbenko, Gregory Korchemsky and Didina Şerban for interesting discussions and Lucia Cordova for collaboration on related subjects. We also thank the participants at the KITP program "Chaos and Order: from Strongly Correlated Systems to Black Holes," the CERN theory institute "Advances in Quantum Field Theory" and the Simons center workshop "TTbar and Other Solvable Deformations of Quantum Field Theories" for stimulating discussions. This research was supported in part by the National Science Foundation under grant no. NSF PHY-1748958, as well as by the ERC Starting Grant 679278 Emergent-BH and the Swedish Research Council grant number 2015-05333. T.A. also acknowledges supported from the Natural Sciences and Engineering Research Council of Canada, grant 376206 from the Simons Foundation, as well as the Delta ITP consortium, a program of the Netherlands Organisation for Scientific Research (NWO) that is funded by the Dutch Ministry of Education, Culture and Science (OCW).

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
