# Peer review of "A general definition of $JT_a$ -- deformed QFTs"

_SciPost Physics, doi:SciPost Phys. 10, 096 (2021)_

## Round 2 · Referee Report · Anonymous (Referee 2) · 2021-4-8

Report

The authors approach the analysis of the JT deformation from the path integral point of view. In this framework they derive the partition function as a path integral transform and use this result to derive an equation for the spectrum. Then they solve this equation finding results matching the literature. Moreover they derive the dressing factor for the S-matrix, again, agreeing with past analyses.
The paper is clear and well written and I suggest it for publication on Sci-Post.

---

## Round 2 · Referee Report · Anonymous (Referee 1) · 2021-4-8

Report

This paper gives a nice account of a variety of approaches to the JT deformation, in analogy to he various methods that have been used to study TTbar. In particular a thorough account is given of a path integral definition of the deformation. The paper is well written and also serves as a useful survey of the state of the art for the TTbar deformation. From the point of view of uniqueness, it would have been useful, but not necessary, to have had some discussion about boundary conditions on the pde (3.13) and to what extent they make its solution unique. (This is a problem for TTbar because the rhs is not elliptic). Also whether the spectral decomposition, which satisfies the pde term-by-term, in fact converges uniformly.

---

## Editorial Decision

published